# A Novel GNN Framework Integrating Neuroimaging and Behavioral Information to Understand Adolescent Psychiatric Disorders

**Weifeng Yu**[1]                                                    DAG9WJ@VIRGINIA.EDU
**Gang Qu**[2]                                                          GQU1@TULANE.EDU
**Young-geun Kim**[3]                                               KIMYO145@MSU.EDU
**Lei Xu**[1]                                                            LX557@NYU.EDU
**Aiying Zhang**[1]                                                  XSA5DB@VIRGINIA.EDU

[1] *School of Data Science, University of Virginia, Charlottesville, VA*

[2] *Department of Biomedical Engineering, Tulane University, New Orleans, LA*

[3] *Department of Statistics and Probability, Michigan State University, East Lansing, MI*

**Editors:** Accepted for publication at MIDL 2025

## Abstract

Functional connectivity (FC) is widely used to study various psychiatric disorders, but its consistency is often undermined by significant inter-subject variability. While these differences can be reflected in behavioral characteristics, few studies have combined them with FC. To this end, we propose a novel graph learning framework that enhances the differentiation of psychiatric disorders by integrating FC and behavioral characteristics. Additionally, we apply Grad-CAM to enhance model interpretability by identifying key regions of interest involved in distinguishing individuals with psychiatric disorders from healthy controls. Experiments with the Adolescent Brain Cognitive Development dataset highlighted two critical insights: the thalamus and specific ROIs within the somatomotor and cingulo-opercular networks play a critical role for identifying psychiatric disorders. Additionally, visualization of latent representations demonstrated that individuals with externalizing disorders, specifically Attention Deficit Hyperactivity Disorder and Oppositional Defiant Disorder, can be distinguished from healthy controls. These findings underscore the utility of our graph learning framework for identifying psychiatric disorders and suggest its promise for improving diagnostic accuracy. Our code is available at https://github.com/elleryyu/BEG-GAE.

**Keywords:** fMRI, adolescent psychiatric disorder, neurobehavior, graph autoencoder, interpretability.

## 1. Introduction

Late childhood and early adolescence are critical stages for brain functional development, often accompanied by the onset and development of multiple psychiatric problems, including anxiety disorders (ANX) (Siegel and Dickstein, 2011), obsessive–compulsive disorder (OCD), oppositional defiant disorder (ODD) (Ghosh et al., 2017), conduct disorder (CD) (Fairchild et al., 2019; Stein et al., 2019), and attention-deficit hyperactivity disorder (ADHD) (Swanson et al., 1998; Sun et al., 2022), that affect cognitive development, social functioning, and overall quality of life, potentially leading to long-term impairments and heightened risk for persistent psychiatric disorders in adulthood (Costello et al., 2003). It

is crucial to understand the underlying neurobehavioral mechanisms of these disorders at early stages and to identify biomarkers that could potentially inform the development of effective prevention and intervention strategies.

Functional magnetic resonance imaging (fMRI) offers a non-invasive, high-resolution method for capturing brain activity by detecting fluctuations in blood-oxygenation-level-dependent (BOLD) signals. Though BOLD signals provide substantial information about neural activity, their temporal characteristics make it challenging to reveal synchronous activity for inter-regional brain communication (Yan et al., 2022; Wang et al., 2024b). To alleviate this issue, functional connectivity (FC) (Smitha et al., 2017), estimated as the temporal association between different regions of interest (ROIs) derived from the fMRI time-series data, has become a crucial tool for phenotype association study (Orlichenko et al., 2022) and psychiatric disorders research (Zhou et al., 2020; Zhang et al., 2019).

Graph Neural Networks (GNNs) are powerful paradigms for embedding graph-structured data, with the capability to integrate complex brain networks (Zhang et al., 2022; Wang et al., 2023; Zhu et al., 2022). This capability is particularly valuable for neuroimaging studies, as it facilitates comprehensive analysis of brain structures and the functional interactions between ROIs. Prior studies have demonstrated that population-level graph representations are effective for tasks such as demographic classification, brain cognition (Qu et al., 2021a; Xiao et al., 2020), and development (Xiao et al., 2022; Chen et al., 2024) studies. However, psychiatric disorder classification is inherently more challenging than cognitive ability classification due to high inter-subject variability and FC heterogeneity (Langhammer et al., 2024; Wang et al., 2024a), which often obscure condition-specific patterns. Incorporating relationship at population level can capture a range of factors beyond individual FC, thereby enabling the construction of a richer network of relationships among participants. By integrating this information, the model gains additional context that refines its capacity to discern and classify distinct disorders. To this end, we propose **B**ehavioral **E**dge **G**eneration **G**raph **A**uto**E**ncoder (**BEG-GAE**), a novel GNN framework that integrates relevant behavioral characteristics with FC data to enhance the brain network representation and to identify key brain regions underlying psychiatric disorders. In this approach, node features are derived from FC data, while edge features encode behavioral characteristics, enabling the model to capture subtle connectivity changes linked to psychiatric disorders. To further enhance the interpretability of the model, we adopt the gradient-weighted class activation mapping (Grad-CAM) (Selvaraju et al., 2017; Qu et al., 2021b) to highlight the ROIs that are most critical for classification of psychiatric disorders.

## 2. Methodology

As shown in Figure 1, the BEG-GAE consists of: 1) **Node (i.e., subject) feature extraction**: For each subject in the population graph, the node features are defined as graph embeddings generated from the subject's FC using $GAE_{FC}$; 2) **Edge generation**: A weighted adjacency matrix is constructed to capture subject similarities based on behavioral characteristics; 3) **Latent representation**: Latent features are extracted from the population graph using population graph GAE ($GAE_{pop}$) to help distinguish psychiatric disorders; 4) **Performance evaluation and biomarker identification**: Classification is performed for both validation and feature analysis using logistic regression and Grad-CAM.

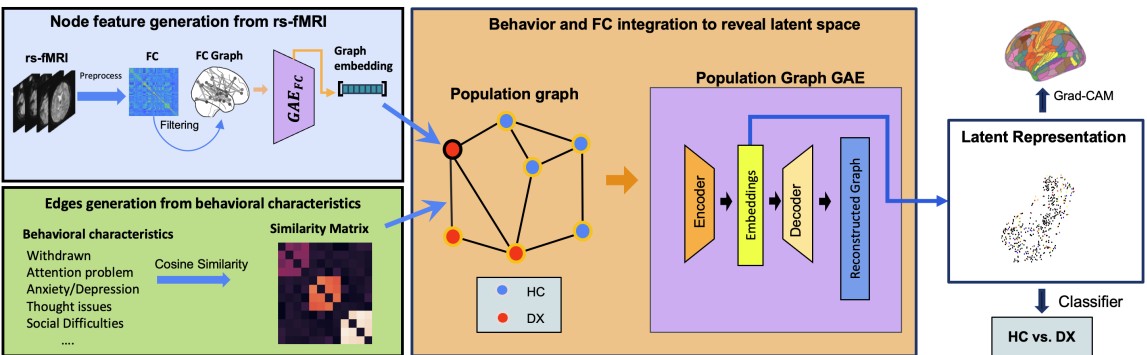

Figure 1: Schematic diagram of the BEG-GAE.

## 2.1. Embedding Extraction using Graph Autoencoder

Latent features are derived from Graph Autoencoder (GAE) (Kipf and Welling, 2016). For an input graph, all node features are concatenated into the feature matrix $\boldsymbol{X} \in \mathbb{R}^{n \times d}$, where $n$ is the number of nodes and $d$ is the feature dimensionality. During the encoding phase, the graph convolutional layer processes $\boldsymbol{X}$ and produces a latent representation $\boldsymbol{H}$:

$$\boldsymbol{H} = \sigma\big(\tilde{\boldsymbol{D}}^{-\frac{1}{2}}\,\tilde{\boldsymbol{A}}\,\tilde{\boldsymbol{D}}^{-\frac{1}{2}}\,\boldsymbol{X}\,\boldsymbol{W}\big) \quad \text{and} \quad \tilde{\boldsymbol{A}} = \boldsymbol{A} + \boldsymbol{I},$$

where $\tilde{\boldsymbol{A}}$ is the adjacency matrix with self-loop ($\boldsymbol{I}$ is the identity matrix), $\tilde{\boldsymbol{D}}$ is the degree matrix of $\tilde{\boldsymbol{A}}$, $\boldsymbol{W}$ is the learnable weight matrix, and $\sigma(\cdot)$ is the nonlinear activation function.

In the decoding step, the latent representation $\boldsymbol{H}$ is used to reconstruct an approximation $\boldsymbol{X}'$ of the original feature matrix:

$$\boldsymbol{X}' = \sigma\big(\tilde{\boldsymbol{D}}^{-\frac{1}{2}}\,\tilde{\boldsymbol{A}}\,\tilde{\boldsymbol{D}}^{-\frac{1}{2}}\,\boldsymbol{H}\,\boldsymbol{W}'\big),$$

where $\boldsymbol{W}'$ is the reconstruction weight matrix. The optimization objective minimizes the Mean Squared Error (MSE) loss between $\boldsymbol{X}'_{\boldsymbol{i}}$ and $\boldsymbol{X}_{\boldsymbol{i}}$, defined as:

$$\mathcal{L}_{\mathrm{MSE}} = \frac{1}{nd}\sum_{i=1}^{n}\sum_{j=1}^{d}(\boldsymbol{X}_{ij} - \boldsymbol{X}'_{ij})^2,$$

which encourages the model to learn embeddings that retain essential information of inputs.

## 2.2. Population Graph Generation

A population graph integrates all individuals, where each edge represents connections between a pair of subjects. The embeddings of each individual FC using $GAE_{FC}$ serve as the node features in the population graph. We then construct edges of the population graph based on the cosine similarity between behavioral score vectors associated with each subject. For subjects $i$ and $j$ with behavioral score vectors $\boldsymbol{b}_i$ and $\boldsymbol{b}_j$, the cosine similarity is computed as $\mathcal{S}_{ij} = \frac{\boldsymbol{b}_i^{\top}\boldsymbol{b}_j}{\|\boldsymbol{b}_i\|\|\boldsymbol{b}_j\|}$. Connections are only established when the similarity values exceed a predefined threshold, thus controlling the sparsity of the graph and ensuring that the

edges accurately reflect substantive feature similarities. In addition, we explore alternative approach of edge generation using a Euclidean distance-based approach for the population graph, as presented in Table 8 in Appendix F. The results shows that embeddings derived from graphs constructed with cosine similarity achieve performance levels comparable to those built using Euclidean distance.

### 2.3. Model Interpretability with Grad-CAM

Grad-CAM is applied for model interpretability, leveraging the gradient information to compute the importance of each node with respect to the predicted class scores. Specifically, it computes the gradients of the predicted class score $\boldsymbol{y}^c$ with respect to the node embeddings $\boldsymbol{h}^k$ of a graph convolutional layer. The gradient $\boldsymbol{\alpha}_k^c$ for each node embedding $k$ with respect to class $c$ is calculated as $\boldsymbol{\alpha}_k^c = \frac{1}{Z} \sum_i \frac{\partial \boldsymbol{y}^c}{\partial \boldsymbol{h}_i^k}$, where $Z$ is the number of nodes in the layer, and $\frac{\partial \boldsymbol{y}^c}{\partial \boldsymbol{h}_i^k}$ denotes the partial derivative of the score $\boldsymbol{y}^c$ with respect to each node $i$ in the embedding $\boldsymbol{h}^k$. The Grad-CAM heatmap $\boldsymbol{L}^c$ is then generated, followed by a ReLU activation to ensure only positive contributions are retained: $\boldsymbol{L}^c = \text{ReLU}\left(\sum_k \boldsymbol{\alpha}_k^c \boldsymbol{h}^k\right)$. This method enables precise identification of ROIs' contributions to the model decisions, enhancing interpretability by visually identifying key features.

## 3. Experiment and Result

### 3.1. Datasets

We investigated subjects from the University of Utah (UTAH) site of the Adolescent Brain Cognitive Development (ABCD) study, which is designed to explore brain development and mental health for children aged 9–10. Resting-state fMRI (rs-fMRI) and behaviors related to five primary psychiatric disorders were explored, including ANX, OCD, ADHD, ODD, and CD. Participants with less prevalent conditions were excluded, resulting in a final sample of 440 participants (188 female, 252 male, 334 healthy controls (HC), 106 all diagnosed disorders (DX), as shown in Figure 2).

Preprocessed rs-fMRI data from the ABCD study were analyzed following the standardized ABCD pipeline,including motion correction, B0 distortion correction, and gradient nonlinearity adjustments (Hagler Jr et al., 2019). We extracted 379 ROIs using the Glasser atlas (Glasser et al., 2016) for cortical parcellations and the Aseg atlas (Fischl et al., 2002) for subcortical parcellations. Behavioral characteristics were evaluated using the Child Behavior Checklist (CBCL) (Thompson et al., 2019), which measures children's behavioral and emotional functioning across various domains (e.g., withdrawal, somatic complaints, attention problem). In our edge construction, we incorporated all available syndrome scales.

### 3.2. Experimental Setup

To validate the effectiveness of various latent representations in distinguishing between HC and disorder groups, we employed two principal experimental approaches: t-SNE visualization (Van der Maaten and Hinton, 2008) and binary classification tasks. t-SNE visualization

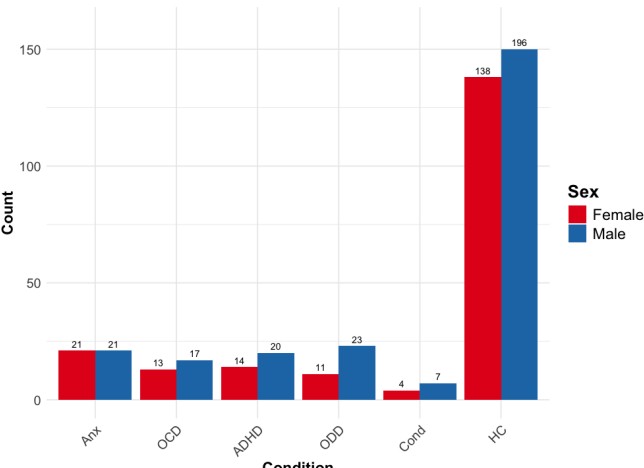

Figure 2: Diagnostic distribution by sex of subjects included in the analysis

was employed to demonstrate the separation of categories within the latent space. For quantitative evaluation, binary classification tasks were conducted using logistic regression with $L1$ regularization (LASSO regression). Each classification task incorporated label balancing and 5-fold cross-validation to ensure robust and unbiased evaluation. The performance of the models was assessed using four metrics: accuracy, F1-scores, recall, and area under the receiver operating characteristic curve (AUC). The classification tasks included HC vs. internalizing disorders (i.e., ANX, OCD), HC vs. externalizing disorders (i.e., ADHD, ODD, CD), and HC vs. DX. Building on these evaluations, we employed pairwise t-tests to statistically assess model performance difference. Metrics derived from the BEG-GAE model served as the baseline for comparison against other approaches.

In addition, we evaluated model generalizability via cross-site validation using two sites of ABCD. The UTAH site (used in all experiments) and the Yale University (YALE) site (used exclusively for cross-site validation) were selected, which have similar labels distribution of the whole dataset. Models were trained on one site and validated on the other, with performance assessed using accuracy, F1-score, recall, and AUC.

Three baseline approaches were compared against the proposed method, progressing from simple raw features to unsupervised learning, and finally to a graph-based approach.

- **Functional Connectivity Only (Flattened FC):** Utilizes raw flattened FC features directly, without embedding.
- **Multilayer Perceptron Autoencoder (AE+FC):** Utilizes an MLP-based autoencoder to flattened FC matrices to learn latent embeddings.
- **Graph Autoencoder (GAE+FC):** Utilizes a Graph Autoencoder to flattened FC matrices to capture the underlying graph structure of brain connectivity.

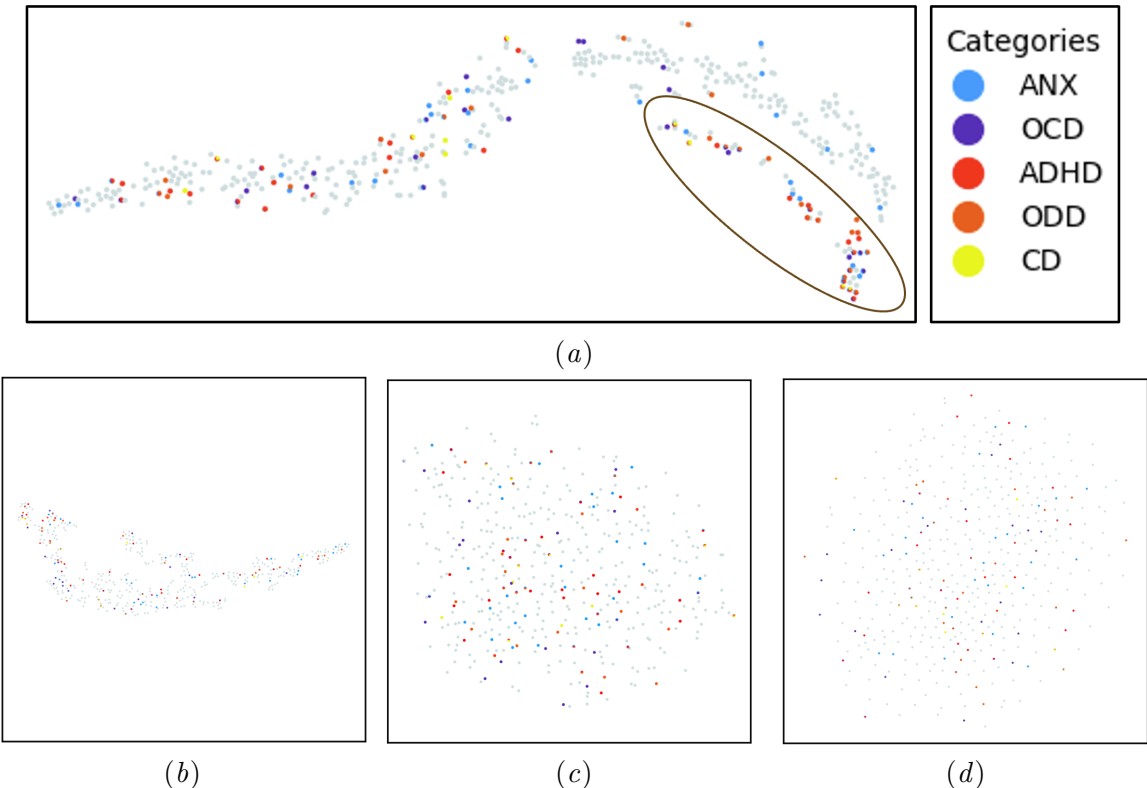

Figure 3: Comparison of t-SNE Dimensionality Reduction Across Different Frameworks: (a) BEG-GAE (**ours**); (b) GAE+FC; (c) MLP Autoencoder+FC; (d) Flattened FC.

### 3.3. Experimental Result

**Distribution of Multilabel Embeddings:** As shown in Figure 3, embeddings generated by other competing methods show minimal separation between psychiatric disorders and HC, with significant overlap in the latent space. Figure 3(a) illustrates that externalizing disorders (i.e., ADHD and ODD) are more densely clustered within the annotated area compared to non-externalizing disorders. In contrast, a larger proportion of ANX and OCD cases tend to lie in the region dominated by healthy controls. To quantify the quality of the generated embeddings, we computed the Calinski-Harabasz score (Caliński and Harabasz, 1974), where a higher value indicates better clustering performance. Our embeddings showed significantly higher scores (in appendix D) than those from other frameworks, indicating that ours learned embeddings that were more clustered according to the existence of externalizing disorders.

**Binary Classification:** As shown in Table 1, BEG-GAE outperformed all other approaches. This indicates that combining behavioral scores with FC yields a more comprehensive representation, enhancing the discriminative capacity of our model, effectively differentiating HCs from those with psychiatric disorders. Additionally, the model achieved better performance in distinguishing the externalizing group from HC compared to the internalizing versus HC classification. Notably, t-tests conducted on the performance met-

| HC vs. Internalizing Disorders | | | | | | | | |
|---|---|---|---|---|---|---|---|---|
| **Framework** | **Accuracy** | **p-value** | **F1** | **p-value** | **Recall** | **p-value** | **AUC** | **p-value** |
| Flattened FC | 0.52 (0.03) | 0.08 | 0.54 (0.04) | 0.38 | 0.57 (0.09) | 0.76 | 0.54 (0.05) | 0.72 |
| AE+FC | 0.52 (0.16) | 0.44 | 0.50 (0.21) | 0.58 | 0.51 (0.24) | 0.80 | **0.60 (0.17)** | 0.83 |
| GAE+FC | 0.56 (0.09) | 0.33 | 0.56 (0.10) | 0.66 | **0.58 (0.14)** | 0.50 | 0.57 (0.13) | 0.96 |
| **BEG-GAE (Ours)** | **0.62 (0.09)** | - | **0.59 (0.11)** | - | 0.56 (0.13) | - | 0.57 (0.12) | - |
| HC vs. Externalizing Disorders | | | | | | | | |
| **Framework** | **Accuracy** | **p-value** | **F1** | **p-value** | **Recall** | **p-value** | **AUC** | **p-value** |
| Flattened FC | 0.56 (0.08) | 0.001 | 0.55 (0.11) | 0.002 | 0.54 (0.12) | 0.035 | 0.61 (0.09) | 0.0009 |
| AE+FC | 0.52 (0.05) | 0.0008 | 0.53 (0.03) | 0.004 | 0.54 (0.07) | 0.05 | 0.55 (0.07) | 0.001 |
| GAE+FC | 0.55 (0.13) | 0.04 | 0.56 (0.11) | 0.04 | 0.56 (0.10) | 0.08 | 0.51 (0.19) | 0.02 |
| **BEG-GAE (Ours)** | **0.79 (0.07)** | - | **0.78 (0.08)** | - | **0.79 (0.13)** | - | **0.82 (0.09)** | - |
| HC vs. DX | | | | | | | | |
| **Framework** | **Accuracy** | **p-value** | **F1** | **p-value** | **Recall** | **p-value** | **AUC** | **p-value** |
| Flattened FC | 0.53 (0.05) | 0.001 | 0.52 (0.08) | 0.005 | 0.52 (0.14) | 0.021 | 0.53 (0.05) | 0.004 |
| AE+FC | 0.55 (0.09) | 0.05 | 0.55 (0.08) | 0.028 | 0.57 (0.11) | 0.019 | 0.57 (0.10) | 0.077 |
| GAE+FC | 0.50 (0.03) | 0.001 | 0.48 (0.07) | 0.001 | 0.47 (0.11) | 0.003 | 0.56 (0.02) | 0.01 |
| **BEG-GAE (Ours)** | **0.73 (0.06)** | - | **0.74 (0.06)** | - | **0.78 (0.10)** | - | **0.75 (0.07)** | - |

\* The mean and standard deviation (in parentheses) are reported.

Table 1: Classification Performance Comparison

rics revealed statistically significant differences for performance comparisons between our method and competing approaches, except for the HC versus internalizing disorders classification, where the results were not significant. We discuss implications of the non-significant t-test results in the discussion section. Additionally, we extend the baseline model by incorporating both FC data and CBCL scores as inputs to the framework. The results presented in Table 2 in Appendix A indicate that the embeddings generated by the BEG-GAE model achieve marginally better performance than those produced by the AE and demonstrate comparable performance to those generated by the GAE.

**Cross-site Validation:** As shown in Table 6 and Table 7 (in Appendix E), the BEG-GAE model trained on the UTAH site performed similarly when validated on YALE site as the model trained on YALE site did on its own subjects, demonstrating the model's generalizability across sites.

**Identifying Key ROIs Relevant to Psychiatric Disorders:** As shown in Figure 4, Grad-CAM analysis identifies several ROIs that help differentiate between HC and DX groups. Notably, the thalamus stands out for its role in sensory relay and regulation of consciousness, which has been linked to ADHD (Ivanov et al., 2010). Additionally, regions within the somatomotor network, essential for voluntary motor control and coordination, and the cingulo-opercular network, crucial for cognitive control and emotional regulation, show strong associations with psychiatric disorders, with particularly notable links to ADHD (Norman et al., 2021; Wang et al., 2022). These findings suggest that these networks and regions potentially play a significant role in understanding psychiatric disorders.

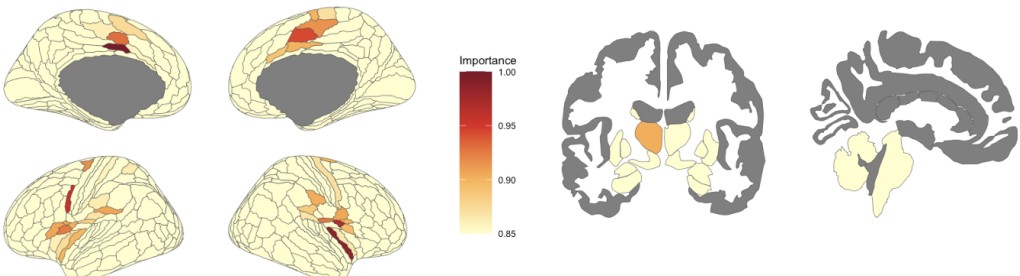

Figure 4: Grad-CAM–identified regions of interest (ROIs) related to psychiatric disorders. Left: Cortical regions; Right: Subcortical regions. Node importance values ranging from 0 to 1, representing each ROI's importance in distinguishing psychiatric disorders. In each figure, non-target regions are shaded in gray.

## 4. Discussion

### 4.1. Challenges in Differentiating Internalizing Disorders from Healthy Control

The substantial overlap between the ANX group and healthy controls (HC), as illustrated in Figure 3(a), is indicative of the poor classification performance observed in distinguishing HC from internalizing disorders. This overlap may also explain why our model failed to capture the involvement of fronto-parietal network regions, which are commonly associated with anxiety disorders (Ma et al., 2019).

To further investigate these issues, we conducted a detailed examination of our dataset. We discovered a notable imbalance within the anxiety disorder group, which comprised 42 subjects, 36 of whom were diagnosed with Specific Phobia (SPH)—a subtype of ANX that exhibits significant variability in neural signals (Ipser et al., 2013) and is challenging to differentiate from HC. This imbalance likely contributed to the suboptimal classification outcomes for internalizing disorders.

To address this issue, subjects diagnosed exclusively with Specific Phobia were excluded from the analysis. As shown in Table 3 in Appendix B, this exclusion improved the classification performance for Internalizing vs. HC, with the remaining 6 ANX subjects being correctly classified.

### 4.2. Limitations and Future Directions

Though we conducted cross-site validation and employed alternative sampling strategies for classification (See Appendix C) to identify our model exhibits relative generalizability, a larger sample size will further improve the generalizability of our model. While pooling data from multiple sites can increase sample size and improve statistical power, it introduces additional variability that risks obscuring the biological or functional patterns of interest. In future research, we plan to apply site-effect removal techniques, such as ComBat (Yu et al., 2018), to harmonize multi-site data and mitigate scanner-related variability. This approach will allow us to utilize more samples across different sites, thereby facilitating more

generalizable findings and broader applicability, and enabling the identification of specific brain regions that differentiate distinct psychiatric disorders from HC.

Multimodal fusion, as demonstrated in prior research, has been shown to enhance the richness and interpretability of learned representations across applications such as intelligence (Qu et al., 2024), sex classification (Patel et al., 2024), and brain cognition (Hu et al., 2021). However, this study is centered on FC, which provides valuable insights into neural interactions but overlooks other critical dimensions of brain organization. To address this limitation, future work will incorporate additional modalities, such as structural MRI (sMRI) and diffusion tensor imaging (DTI). By integrating these modalities with fMRI, the resulting graph representations are expected to capture complementary and diverse features of brain organization, thereby enriching the representation space and advancing our understanding of complex neural patterns.

## 5. Conclusion

We introduce BEG-GAE, an innovative framework that combines resting-state fMRI data with behavioral characteristics to advance the representation of psychiatric disorders. Our findings reveal that the BEG-GAE model generates representations that surpass traditional methods, including Autoencoders, Graph Autoencoders (GAEs), and raw functional connectivity features. Additionally, our analysis identifies key brain regions, particularly within the somatomotor and cingulo-opercular networks, as critical for classifying psychiatric disorders. These results underscore the potential of BEG-GAE to improve psychiatric diagnostics in late childhood and early adolescence by elucidating the intricate associations between brain connectivity and psychiatric disorders.

## 6. Acknowledgments

This work is partly supported by the UVA Brain Institute Transformative Neuroscience Pilot Grants. Data used in this study were obtained from the Adolescent Brain Cognitive Development[SM] (ABCD) Study (https://abcdstudy.org), held in the NIMH Data Archive (NDA).

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

## Appendix A. Classification Performance Comparison with Different Input for Embeddings Extraction

| Framework | Accuracy | F1 | Recall | AUC |
|---|---|---|---|---|
| **HC vs. Internalizing Disorders** | | | | |
| Raw Features | 0.62 (0.05) | 0.63 (0.07) | 0.65 (0.12) | 0.69 (0.03) |
| AE | 0.59 (0.10) | 0.59 (0.11) | 0.59 (0.15) | 0.58 (0.10) |
| GAE | 0.73 (0.04) | 0.70 (0.06) | 0.64 (0.12) | 0.79 (0.08) |
| **BEG-GAE** | 0.62 (0.09) | 0.59 (0.11) | 0.56 (0.13) | 0.57 (0.12) |
| **HC vs. Externalizing Disorders** | | | | |
| Raw Features | 0.69 (0.04) | 0.67 (0.05) | 0.64 (0.09) | 0.81 (0.08) |
| AE | 0.57 (0.07) | 0.58 (0.08) | 0.59 (0.11) | 0.59 (0.10) |
| GAE | 0.80 (0.10) | 0.80 (0.11) | 0.77 (0.11) | 0.87 (0.08) |
| **BEG-GAE** | 0.79 (0.07) | 0.78 (0.08) | 0.79 (0.13) | 0.82 (0.09) |
| **HC vs. DX** | | | | |
| Raw Features | 0.67 (0.04) | 0.67 (0.06) | 0.67 (0.09) | 0.75 (0.02) |
| AE | 0.70 (0.04) | 0.72 (0.04) | 0.76 (0.10) | 0.76 (0.03) |
| GAE | 0.74 (0.02) | 0.75 (0.02) | 0.79 (0.05) | 0.81 (0.05) |
| **BEG-GAE** | 0.73 (0.06) | 0.74 (0.06) | 0.78 (0.10) | 0.75 (0.07) |

Table 2: Classification Performance: BEG-GAE vs. Baseline Models with Additional CBCL Scores. The mean and standard deviation (in parentheses) are reported.

## Appendix B. Subgroup Analysis

In the current dataset, there are 42 subjects with ANX, 36 of whom are diagnosed with Specific Phobia, which has led to poor classification performance in internalizing disorders.

To address this issue, we excluded subjects with Specific Phobia only. As shown in Table 3, the classification result for Internalizing vs. HC is improved, indicating that subjects with only Specific Phobia negatively impact classification results.

| Classification Task | Accuracy | F1-score | Recall | AUC |
|---|---|---|---|---|
| Internalizing vs. HC (Include Specific Phobia) | 0.62 (0.09) | 0.59 (0.11) | 0.56 (0.13) | 0.57 (0.12) |
| Internalizing vs HC (Exclude Specific Phobia) | 0.66 (0.14) | 0.64 (0.14) | 0.60 (0.14) | 0.67 (0.16) |
| Externalizing vs HC | 0.79 (0.07) | 0.78 (0.08) | 0.79 (0.13) | 0.82 (0.09) |
| HC vs DX | 0.73 (0.04) | 0.74 (0.04) | 0.75 (0.10) | 0.74 (0.05) |
| HC vs OCD | 0.65 (0.06) | 0.67 (0.07) | 0.73 (0.13) | 0.63 (0.17) |
| HC vs ADHD | 0.80 (0.09) | 0.79 (0.09) | 0.79 (0.08) | 0.80 (0.15) |
| HC vs ODD | 0.81 (0.08) | 0.80 (0.09) | 0.82 (0.15) | 0.84 (0.07) |
| HC vs CD | 0.73 (0.28) | 0.78 (0.23) | 0.90 (0.20) | 0.87 (0.19) |

Table 3: Logistic Regression Results for Subgroup Comparisons.

## Appendix C. Impact of Sampling Strategies on HC vs. DX Classification Performance

Although we employed label balancing strategies for LASSO regression in the Binary Classification task described in Section 3, it remains crucial to evaluate whether the imbalance in data across

categories affected the results. To further investigate this, we re-evaluated the HC vs. DX classification by applying Synthetic Minority Over-sampling Technique (SMOTE) (Chawla et al., 2002) and Weighted Cross Entropy (WCE) (Aurelio et al., 2019) in addition to downsampling. These experiments were conducted under the same experimental settings as the original Binary Classification task. As shown in Table 4, BEG-GAE demonstrates stable performance across different sampling methods, whereas AE and GAE exhibit greater variability.

| Framework | Sampling Strategies | Accuracy | F1-score | Recall | AUC |
|---|---|---|---|---|---|
| AE+FC | SMOTE | 0.73 (0.06) | 0.71 (0.06) | 0.66 (0.04) | 0.75 (0.05) |
| AE+FC | WCE | 0.59 (0.05) | 0.71 (0.04) | 0.64 (0.05) | 0.57 (0.08) |
| AE+FC | Label balancing | 0.55 (0.09) | 0.55 (0.08) | 0.57 (0.11) | 0.57 (0.10) |
| GAE+FC | SMOTE | 0.67 (0.03) | 0.64 (0.02) | 0.59 (0.03) | 0.71 (0.02) |
| GAE+FC | WCE | 0.57 (0.04) | 0.67 (0.04) | 0.59 (0.05) | 0.59 (0.04) |
| GAE+FC | Label balancing | 0.50 (0.03) | 0.48 (0.07) | 0.47 (0.11) | 0.56 (0.02) |
| BEG-GAE | SMOTE | 0.76 (0.02) | 0.76 (0.03) | 0.76 (0.07) | 0.81 (0.03) |
| BEG-GAE | WCE | 0.75 (0.06) | 0.82 (0.05) | 0.77 (0.06) | 0.75 (0.09) |
| BEG-GAE | Label balancing | 0.73 (0.06) | 0.74 (0.06) | 0.78 (0.10) | 0.75 (0.07) |

Table 4: HC vs. DX Classification Performance Using Different Sampling Strategies

## Appendix D. Clustering Evaluation for Generated Embeddings

To further evaluate clustering performance, we compute the Calinski-Harabasz (CH) scores and analyze the clustering results for HC and DX groups using the same embeddings as those presented in the t-SNE visualization. Specifically, as shown in the **CH Scores (Cluster 0: HC; Cluster 1: DX)** column of Table 5, the clusters generated by BEG-GAE demonstrate superior structure compared to those derived from baseline embeddings.

| Framework | CH Scores | CH Scores after SMOTE |
|---|---|---|
| **BEG-GAE** | 3.64 | 12.10 |
| Flattened FC | 1.14 | **N/A** |
| AE+FC | 2.04 | 6.22 |
| GAE+FC | 0.39 | 1.79 |

Table 5: Calinski-Harabasz Score for the UTAH site when we use HC as Cluster 0 and DX as Cluster 1.

## Appendix E.  Cross-Site Validation for Assessing Model Generalizability

| Framework | Accuracy | | F1 | | Recall | | AUC | |
|---|---|---|---|---|---|---|---|---|
| | Site A | Site B | Site A | Site B | Site A | Site B | Site A | Site B |
| **HC vs. Internalizing Disorders** | | | | | | | | |
| Flattened FC | 0.52 (0.03) | 0.45 (0.08) | 0.54 (0.04) | 0.47 (0.12) | 0.57 (0.09) | 0.51 (0.17) | 0.54 (0.05) | 0.43 (0.08) |
| AE+FC | 0.52 (0.16) | 0.52 (0.06) | 0.50 (0.21) | 0.50 (0.13) | 0.51 (0.24) | 0.53 (0.25) | 0.60 (0.17) | 0.47 (0.06) |
| GAE+FC | 0.56 (0.09) | 0.43 (0.10) | 0.56 (0.10) | 0.44 (0.12) | 0.58 (0.14) | 0.46 (0.15) | 0.57 (0.13) | 0.45 (0.15) |
| BEG-GAE | 0.62 (0.09) | 0.69 (0.09) | 0.59 (0.11) | 0.65 (0.11) | 0.56 (0.13) | 0.59 (0.15) | 0.57 (0.12) | 0.70 (0.13) |
| **HC vs. Externalizing Disorders** | | | | | | | | |
| Flattened FC | 0.56 (0.08) | 0.46 (0.08) | 0.55 (0.11) | 0.48 (0.16) | 0.54 (0.12) | 0.54 (0.27) | 0.61 (0.09) | 0.44 (0.15) |
| AE+FC | 0.52 (0.05) | 0.31 (0.07) | 0.53 (0.03) | 0.32 (0.06) | 0.54 (0.07) | 0.33 (0.07) | 0.55 (0.07) | 0.27 (0.10) |
| GAE+FC | 0.55 (0.13) | 0.46 (0.13) | 0.56 (0.11) | 0.48 (0.15) | 0.56 (0.10) | 0.51 (0.18) | 0.51 (0.19) | 0.43 (0.11) |
| BEG-GAE | 0.79 (0.07) | 0.83 (0.10) | 0.78 (0.08) | 0.85 (0.07) | 0.79 (0.13) | 0.89 (0.09) | 0.82 (0.09) | 0.82 (0.11) |
| **HC vs. DX** | | | | | | | | |
| Flattened FC | 0.53 (0.05) | 0.43 (0.10) | 0.52 (0.08) | 0.43 (0.12) | 0.52 (0.14) | 0.44 (0.14) | 0.53 (0.05) | 0.37 (0.10) |
| AE+FC | 0.55 (0.09) | 0.48 (0.13) | 0.55 (0.08) | 0.47 (0.20) | 0.57 (0.11) | 0.50 (0.23) | 0.57 (0.10) | 0.41 (0.15) |
| GAE+FC | 0.50 (0.03) | 0.41 (0.04) | 0.48 (0.07) | 0.38 (0.04) | 0.47 (0.11) | 0.36 (0.06) | 0.56 (0.02) | 0.39 (0.05) |
| BEG-GAE | 0.73 (0.06) | 0.78 (0.08) | 0.74 (0.06) | 0.79 (0.07) | 0.78 (0.10) | 0.84 (0.07) | 0.75 (0.07) | 0.83 (0.06) |

Table 6: Cross-Site Classification Performance: BEG-GAE trained on UTAH site (Site A) and validated on YALE site (Site B)

| Framework | Accuracy | | F1 | | Recall | | AUC | |
|---|---|---|---|---|---|---|---|---|
| | Site A | Site B | Site A | Site B | Site A | Site B | Site A | Site B |
| **HC vs. Internalizing Disorders** | | | | | | | | |
| AE+FC | 0.46 (0.06) | 0.56 (0.06) | 0.44 (0.08) | 0.58 (0.05) | 0.43 (0.11) | 0.61 (0.08) | 0.45 (0.06) | 0.53 (0.10) |
| GAE+FC | 0.57 (0.06) | 0.50 (0.17) | 0.57 (0.06) | 0.49 (0.17) | 0.57 (0.12) | 0.48 (0.18) | 0.55 (0.06) | 0.51 (0.11) |
| BEG-GAE | 0.62 (0.07) | 0.68 (0.05) | 0.61 (0.06) | 0.61 (0.11) | 0.59 (0.03) | 0.53 (0.17) | 0.61 (0.07) | 0.66 (0.09) |
| **HC vs. Externalizing Disorders** | | | | | | | | |
| AE+FC | 0.51 (0.04) | 0.54 (0.15) | 0.54 (0.04) | 0.57 (0.14) | 0.58 (0.08) | 0.63 (0.17) | 0.52 (0.10) | 0.54 (0.15) |
| GAE+FC | 0.51 (0.05) | 0.47 (0.16) | 0.50 (0.07) | 0.45 (0.22) | 0.51 (0.12) | 0.48 (0.29) | 0.53 (0.08) | 0.41 (0.12) |
| BEG-GAE | 0.73 (0.07) | 0.87 (0.05) | 0.72 (0.08) | 0.87 (0.04) | 0.72 (0.13) | 0.85 (0.08) | 0.72 (0.08) | 0.87 (0.08) |
| **HC vs. DX** | | | | | | | | |
| AE+FC | 0.61 (0.03) | 0.44 (0.12) | 0.61 (0.05) | 0.44 (0.12) | 0.61 (0.10) | 0.44 (0.11) | 0.62 (0.15) | 0.48 (0.13) |
| GAE+FC | 0.57 (0.07) | 0.41 (0.12) | 0.56 (0.09) | 0.39 (0.11) | 0.56 (0.11) | 0.37 (0.09) | 0.59 (0.04) | 0.41 (0.11) |
| BEG-GAE | 0.68 (0.05) | 0.77 (0.07) | 0.70 (0.07) | 0.78 (0.06) | 0.76 (0.13) | 0.80 (0.07) | 0.68 (0.08) | 0.75 (0.08) |

Table 7: Cross-Site Classification Performance: BEG-GAE trained on Site B and validated on Site A

## Appendix F. Population Graph Edge Generation in BEG-GAE

In Section 2.2, we introduced edge generation using cosine similarity, setting a predefined threshold of 0.55 to ensure connectivity while maintaining sparsity. Additionally, we explored constructing edges based on Euclidean distance to form a Nearest Neighbor graph. We evaluated the classification performance of the extracted embeddings in this two different method of edge generation. As shown in Table 8, the results show that embeddings extracted from graphs built using cosine similarity achieve a performance level comparable to those constructed with Euclidean distance.

| Task | Measurement | Create | Accuracy | F1 | Recall | AUC |
|---|---|---|---|---|---|---|
| HC vs. Internalizing | Euclidean Distance | Nearest Neighbors | 0.63 (0.11) | 0.62 (0.10) | 0.62 (0.13) | 0.69 (0.12) |
| HC vs. Internalizing | Cosine Similarity | Thresholding | 0.62 (0.09) | 0.59 (0.11) | 0.56 (0.13) | 0.57 (0.12) |
| HC vs. Externalizing | Euclidean Distance | Nearest Neighbors | 0.79 (0.05) | 0.81 (0.04) | 0.85 (0.08) | 0.83 (0.07) |
| HC vs. Externalizing | Cosine Similarity | Thresholding | 0.79 (0.07) | 0.78 (0.08) | 0.79 (0.13) | 0.82 (0.09) |
| HC vs. DX | Euclidean Distance | Nearest Neighbors | 0.71 (0.08) | 0.70 (0.12) | 0.71 (0.18) | 0.77 (0.06) |
| HC vs. DX | Cosine Similarity | Thresholding | 0.73 (0.06) | 0.74 (0.06) | 0.78 (0.10) | 0.75 (0.07) |

Table 8: Performance Comparison of Population Graph Edge Generation in BEG-GAE using Different Measurements and Construction Methods

