# OpenReview forum: "A Novel GNN Framework Integrating Neuroimaging and Behavioral Information to Understand Adolescent Psychiatric Disorders"
_MIDL.io/2025/Conference — MIDL 2025 Poster_

### Official Review · Reviewer_JiJN · 2025-02-13

**Confidence:** 4
**Preliminary Rating:** 4
**Recommendation:** Poster

**Summary:**

This study presents BEG-GAE, a GNN framework that integrates fMRI-based functional connectivity and behavioral data to investigate adolescent psychiatric disorders. It utilizes GAE for node embedding and behavioral similarity to generate edges, with Grad-CAM applied for interpretability. Using the ABCD dataset, t-SNE visualization and binary classification tasks were conducted, revealing that BEG-GAE outperforms baseline methods, particularly in distinguishing externalizing disorders. It also highlights important ROIs like the thalamus, contributing to improved representation and diagnostic accuracy for psychiatric disorders in adolescents.

**Strengths:**

1. The innovative BEG—GAE framework integrates neuroimaging (FC data) and behavioral information. This multimodal approach enhances understanding of psychiatric disorders and could lead to improved diagnostic models.
2. Grad-CAM enhances interpretability, enabling the identification of key brain regions involved in distinguishing psychiatric disorders from healthy controls and aiding in understanding neurobiological mechanisms.
3. The paper features well-designed experiments using the ABCD dataset, employing t-SNE visualization, binary classification tasks, and cross-validation to comprehensively evaluate the model against various baseline methods.
4. The paper is well-structured with accessible scientific language, making it easy to follow. It appropriately addresses prior work and the limitations of previous studies.

**Weaknesses:**

1. The study uses data from a single site, which may introduce unique biases. This limits the findings' applicability to a broader population.
2. The model struggles to differentiate internalizing disorders from healthy controls, indicating a need for refinement to better capture disorder-specific features.

**Detailed Comments:**

1.  The paper notes using behavioral scores from the CBCL to generate edges in the population graph but lacks detail on score selection or weighting. Clarifying whether all syndrome scale scores were equally used or if a subset was chosen based on prior knowledge would enhance reader understanding.

2.  Although Grad-CAM identified key ROIs, providing a deeper analysis of their interactions within psychiatric disorders would improve the paper. Discussing functional connections between the thalamus and other networks could reveal disrupted mechanisms in patients.

3.  The binary classification tasks are imbalanced between healthy controls and diagnosed cases. While label balancing techniques are used, further discussion on how this imbalance affects performance metrics and exploring alternative methods for future research would be valuable.

4.  The predefined threshold for constructing edges based on cosine similarity warrants more explanation. Clarifying how this threshold was determined would aid in reproducing results and understanding model sensitivity.

5.  While the paper compares BEG-GAE with some baselines, comparisons with more recent state-of-the-art approaches in graph-based methods for psychiatric disorder classification would better highlight the framework's unique contributions.

**Justification Of The Preliminary Rating:**

The paper has substantial aspects, particularly its novel neuroimaging and behavioral data integration through the BEG - GAE framework, which enhances understanding adolescent psychiatric disorders. The use of Grad-CAM for interpretability is a significant advantage, allowing for the identification of key brain regions. The experiments are well-designed with proper validation. However, reliance on single-site data limits generalizability, and there's a lack of multimodal integration beyond FC and behavior. Additionally, the classification of internalizing disorders needs improvement. Nonetheless, the paper presents innovative ideas with potential for further development in the field.

**Questions To Address In The Rebuttal:**

1. Authors should clarify plans for multi-site data handling and present pilot results to show effectiveness in reducing variability.

2. Explore alternative classification strategies and conduct subgroup analyses to improve understanding and performance.

**Special Issue:**

No

---

> ### Author Response · Authors · 2025-03-08
>
> We would like to express our heartfelt appreciation to the reviewer JiJN. Your detailed feedback has played a crucial role in improving the clarity and robustness of our work. Below, we provide our point-by-point responses to address each comment.
>
> ---
>
> **[Weakness]**
>
> 1. The study uses data from a single site, which may introduce unique biases. This limits the findings' applicability to a broader population.
>
> **Response:**
> In response to your suggestions, we implemented cross-site validation using data from another subgroup (57 DX, 141 HC). According to the results in Section 3.3, there were no notable performance differences between the training and test sites.
>
> ---
>
> 2. The model struggles to differentiate internalizing disorders from healthy controls, indicating a need for refinement to better capture disorder-specific features.
>
> **Response:**
> Thank you for your valuable suggestion. We further explored this issue as discussed in Section 4.1. Specifically, we observed that excluding Specific Phobia, a subtype of ANX, led to improved classification performance for distinguishing Internalizing from HC. Notably, the remaining six ANX subjects were correctly classified following this adjustment.
>
> ---
>
> **[Comment]**
>
> 1. The paper notes using behavioral scores from the CBCL to generate edges in the population graph but lacks detail on score selection or weighting. Clarifying whether all syndrome scale scores were equally used or if a subset was chosen based on prior knowledge would enhance reader understanding.
>
> **Response:**
> We clarified this in Section 3.1 and highlighted that: "In our edge construction, we incorporated all available syndrome scales."
>
> ---
>
> 2. Although Grad-CAM identified key ROIs, providing a deeper analysis of their interactions within psychiatric disorders would improve the paper. Discussing functional connections between the thalamus and other networks could reveal disrupted mechanisms in patients.
>
> **Response:**
> Thank you for the comment. We clarified this in Experimental Results:
> "Notably, the thalamus stands out for its role in sensory relay and regulation of consciousness, which has been linked to ADHD (Ivanov et al., 2010). Additionally, regions within the somatomotor network, essential for voluntary motor control and coordination, and the cingulo-opercular network, crucial for cognitive control and emotional regulation, show strong associations with psychiatric disorders, with particularly notable links to ADHD (Norman et al., 2021; Wang et al., 2022)."
>
> ---
>
> 3. The binary classification tasks are imbalanced between healthy controls and diagnosed cases. While label balancing techniques are used, further discussion on how this imbalance affects performance metrics and exploring alternative methods for future research would be valuable.
>
> **Response:**
> Thank you for the constructive suggestion. Following your suggestion, we used advanced techniques to alleviate the class-imbalance issue, including SMOTE and Weighted Cross Entropy, and the results showed better performance than label balancing techniques. Details are described in Appendix C.
>
> ---
>
> 4. The predefined threshold for constructing edges based on cosine similarity warrants more explanation. Clarifying how this threshold was determined would aid in reproducing results and understanding model sensitivity.
>
> **Response:**
> Thank you for the suggestion. We described this in Appendix F, "setting a predefined threshold of 0.55 to ensure connectivity while maintaining sparsity."
>
> ---
>
> **[Questions To Address In The Rebuttal]**
>
> 1. Authors should clarify plans for multi-site data handling and present pilot results to show effectiveness in reducing variability.
>
> **Response:**
> We performed cross-site validation to show our approach is applicable to multiple sites. In addition, we plan to further mitigate site effects using ComBat and gather samples across different sites. While we do not have updated results at this stage, this is an essential part of our ongoing work, which we have explicitly outlined as the Further Work in Section 4.2.
>
> ---
>
> 2. Explore alternative classification strategies and conduct subgroup analyses to improve understanding and performance.
>
> **Response:**
> Thank you for your constructive suggestion. Following your advice, we utilized Calinski-Harabasz scores to evaluate the clustering performance of HC and DX-labeled subjects, demonstrating improved results compared to other methods (in Appendix D).
>
>  Additionally, for subgroup analysis, we observed an overall improvement in the internalizing vs. HC classification after removing the Specific Phobia subclass from the ANX group. Under this setting, we conducted a detailed analysis across other subgroups, finding that ODD and ADHD could be well distinguished. The detailed results of these analyses are provided in Appendix B.

---

### Official Review · Reviewer_rjcx · 2025-02-18

**Confidence:** 5
**Preliminary Rating:** 4
**Recommendation:** Poster
**Final Rating:** 4

**Summary:**

The author proposed a novel GNN-based framework, BEG-GAE, that integrates behavioral measurements with functional connectivity to enhance psychiatric disorders classification. The model uses cosine similarity based on behavioral characteristics as edges and uses FC generated using a GAE as node. Grad-CAM is used to visualize latent representation for interpretability.

**Strengths:**

The method presented in the paper is clear. Integrating behavioral data combined with FC via GNN increases the model's accuracy compared to using flattened FC alone.
The paper includes multiple baseline comparisons and statistical testing to validate the effectiveness of their model.
Key ROIS are identified using Grad-CAM that suggest the critical regional differences between psychiatric disorders and healthy controls.

**Weaknesses:**

1. The model is only trained and tested on one site of the ABCD dataset; no cross-site validation was conducted. Within the dataset selected, there are only 106 participants with diagnosed disorders while 334 are healthy controls.

2. Constructing edge similarity with cosine similarity is great, but it could also be supplemented with other similarity measurements, given that the authors already obtained subject-level graph embeddings, which might better capture the highly variable psychiatric disorders.

3. In figure 3, the authors showed a comparison of t-SNE across different frameworks; however, there are no quantitative results showing the competing method provides minimal separation. From a visual perspective, figure 3a and figure 3b show very little differences.

**Detailed Comments:**

Fig 4 has no explanation for the large gray areas shown.

**Justification Of The Final Rating:**

The authors addressed my questions in my initial review and have since explored more datasets and methodologies regarding class imbalance, and similarity measures. However, due to the limitation, the dataset size is small. So I would like to keep the original score.

**Justification Of The Preliminary Rating:**

The paper overall is well written with clear methodology. The paper suggests a new way of combining behavioral measures with FC to predict psychiatric disorders. However, the dataset chosen is relatively small, with only 106 participants across all 5 primary psychiatric disorders.

**Questions To Address In The Rebuttal:**

Have you considered using other representation to consturct edge? Graph embeddings from the autoencoders should provide useful and condensed information related to behavioral conditions.

**Special Issue:**

No

---

> ### Author Response · Authors · 2025-03-08
>
> First and foremost, we would like to express our heartfelt appreciation to the reviewer rjcx's careful reading and constructive comments. We greatly value your insights, which have significantly improved our work. Below, we provide our point-by-point responses to address each comment.
>
> ---
>
> **[Weakness]**
>
> 1. The model is only trained and tested on one site of the ABCD dataset; no cross-site validation was conducted. Within the dataset selected, there are only 106 participants with diagnosed disorders while 334 are healthy controls.
>
> **Response:**
> In response to your suggestions, we performed cross-site validation. Building on the existing site data, we introduced a new site, trained models separately on each site, and then validated them across sites. The cross-validation results proved to be similar, with further details provided in Section 3.2 and 3.3.
>
> ---
>
> 2. Constructing edge similarity with cosine similarity is great, but it could also be supplemented with other similarity measurements, given that the authors already obtained subject-level graph embeddings, which might better capture the highly variable psychiatric disorders.
>
> **Response:**
> Thank you for the constructive suggestion. We included details Question to Rebuttal.
>
> ---
>
> 3. In Figure 3, the authors showed a comparison of t-SNE across different frameworks; however, there are no quantitative results showing the competing method provides minimal separation. From a visual perspective, Figure 3a and Figure 3b show very little differences.
>
> **Response:**
> Thank you for your insightful feedback. Following your suggestion, we employed the Calinski-Harabasz (CH) score to assess clustering performance. Our results indicate that BEG-GAE achieved a CH score of 3.64, which is notably higher than those obtained using raw features (1.14), AE (2.04), and GAE (0.39), as detailed in Appendix D. Additionally, we have revised Figure 3 to present the updated visualization results, which shows the observable differences.
>
> ---
>
> **[Comment]**
>
> 1. Fig 4 has no explanation for the large gray areas shown.
>
> **Response:**
> Thank you for highlighting this. In Figure 4, the left highlights cortical regions, and the right highlights subcortical regions. The gray marks indicate areas not of interest. We clarified this in the caption of Figure 4.
>
> ---
>
> **[Questions To Address In The Rebuttal]**
>
> 1. Have you considered using other representations to construct edges? Graph embeddings from the autoencoders should provide useful and condensed information related to behavioral conditions.
>
> **Response:**
> Yes, we did consider this! We attempted to construct the edges of the population graph using Euclidean distance, which resulted in similar performance compared with the cosine similarity-based approach. Details are shown in Appendix F.

---

### Official Review · Reviewer_GEJo · 2025-02-20

**Confidence:** 4
**Preliminary Rating:** 2
**Final Rating:** 3

**Summary:**

This study introduces BEG-GAE (Behavioral Edge Generation Graph AutoEncoder), a novel graph learning framework designed to differentiate psychiatric disorders and enhance the diagnostic accuracy of adolescent psychiatric conditions. The approach integrates functional connectivity (FC), derived from fMRI, with behavioral characteristics, providing a more comprehensive representation.

In this framework, node features are extracted from FC data, while edge features are informed by behavioral characteristics. After node feature extraction and edge generation, t-SNE visualization of the latent space reveals weak deviations from primary distribution patterns in some cases, compared to baseline methods that use traditional network structures for learning latent FC embeddings (AE, GAE, flattened features).
Quantitative performance results across multiple binary classification tasks on the latent space demonstrate the effectiveness of combining behavioral scores with FC, compared to baseline methods. Furthermore, the Grad-CAM method is applied for model interpretability by highlighting key regions of interest (ROIs) involved in differentiation.

**Strengths:**

- The problematic is well described and the introduction and literature review is complete.
- The paper is well written.
- The methodology about the aggregation of FC data and behavioral data is interesting.

**Weaknesses:**

The goal of the paper is not clear, although building a framework that integrates behavioral characteristics with FC data is interesting, I am not sure to understand what it could be used for - considering that behavioral characteristics directly correlate with pathologies. This might actually be the point that I am struggling to understand.


While the use of Grad-CAM seems to be the main advantage of the proposed model architecture, more results could be reported - Grad-CAM importance for the different pathologies.


I am struggling to understand the relevance of the classification experiments as behavioral information (directly related to labels) are included in the population graph construction, thus there is information leakage here, while comparison methods are not provided with any of the behavioral information.

**Detailed Comments:**

- clarify what is meant by “deviating from the primary distribution” in the t-SNE visualization, consider visually highlighting this distribution by adding annotation like a circle. Also, consider reducing the size of the dots in the  t-SNE visualization because they are overlapping
- “the BEG-GAE model generates representations that surpass traditional methods”: I am struggling here to understand the comparison as the BEG-GAE already includes behavioral characteristics in its construction, in contrast to the “traditional methods” which only rely on FC data. So it seems logical that BEG-GAE better represents psychiatric disorders.
- If I understood correctly, section 2.1 describes the embedding extraction from the GAE applied on the population graph. Maybe the authors could give a different name to the other GAE (the ones for the FC data analysis) because it is confusing (e.g. GAE_FC and GAE_pop) on the figure 1 and in the text.

**Justification Of The Final Rating:**

Thank you to the authors for their replies and revisions. I changed my rating to borderline. Although the approach to fuse FC and behavioural scores for graph learning is interesting, the results of Appendix A makes me wondering about what it brings compared to the GAE + CBCL - justifying my rating. Nevertheless, I have no doubt that this paper could be of interest and generate interesting discussions at MIDL.

**Justification Of The Preliminary Rating:**

I might have missed the main goal of the proposed framework and, to me, its usefulness is not very well described. Nevertheless, the method is interesting and I would be happy to change my rating if the authors clarify their goal and address my comments.

**Questions To Address In The Rebuttal:**

see weaknesses and detailed comments

---

> ### Author Response · Authors · 2025-03-08
>
> First and foremost, we would like to express our sincere gratitude to the reviewer GEJo for the carefully reading and constructive comments. We have made our point-by-point responses.
>
> ---
>
> 1. "The goal of the paper is not clear, although building a framework that integrates behavioral characteristics with FC data is interesting, I am not sure to understand what it could be used for - considering that behavioral characteristics directly correlate with pathologies. This might actually be the point that I am struggling to understand."
>
> **Response:**
> We appreciate the opportunity to clarify our goal and enhance the presentation of our work. Existing works fall short in learning representations of FC data associated with psychiatric disorders, and our goal is to fill this gap by leveraging information from behavioral scores. Furthermore, although various behavioral scores, such as CBCL scores in our work, correlate with pathologies, they are limited as diagnostic tools for psychiatric disorders, and there is a substantial demand for identifying such pathologies using FC data. Our framework may alleviate this issue.
>
> ---
>
> 2. "While the use of Grad-CAM seems to be the main advantage of the proposed model architecture, more results could be reported - Grad-CAM importance for the different pathologies."
>
> **Response:**
> Due to the limited number of samples available for each psychiatric disorder in our current dataset, our analysis is limited to a broad comparison between healthy controls (HC) and the overall diagnostic group (DX), rather than focusing on disorder-specific pathologies. We plan to address site effects and collect additional samples, enabling us to identify more disorder-specific pathologies in future work.
>
> ---
>
> 3. "I am struggling to understand the relevance of the classification experiments as behavioral information (directly related to labels) are included in the population graph construction, thus there is information leakage here, while comparison methods are not provided with any of the behavioral information."
>
> **Response:**
> In the revision, we examined whether our representations retained substantial information in FC associated with psychiatric disorders, and we found no notable information leakage. Specifically, we constructed an oracle classifier by using FC and CBCL scores without dimension reduction to fit a LASSO classifier. The oracle accuracy was 0.67, which was comparable to the accuracy of our method, 0.73. See (HC vs. DX) in Table 2 in Appendix A for details.
>
> ---
>
> 4. "Clarify what is meant by “deviating from the primary distribution” in the t-SNE visualization, consider visually highlighting this distribution by adding annotation like a circle. Also, consider reducing the size of the dots in the t-SNE visualization because they are overlapping."
>
> **Response:**
> Following your suggestion, we updated figures by highlighting Figure 3(a) annotations and reducing the size of the dots. We appreciate your suggestion, which has clarified the visualization.
>
> ---
>
> 5. " 'The BEG-GAE model generates representations that surpass traditional methods': I am struggling here to understand the comparison as the BEG-GAE already includes behavioral characteristics in its construction, in contrast to the “traditional methods” which only rely on FC data. So it seems logical that BEG-GAE better represents psychiatric disorders."
>
> **Response:**
> Thank you for your insightful feedback. We updated our result and illustrated this in the Binary Classifications in Section 3.3.
> We highlighted: "Additionally, we extend the baseline model by incorporating both FC data and CBCL scores as inputs to the framework. The results presented in Table 2 in Appendix A indicate that the embeddings generated by the BEG-GAE model achieve marginally better performance than those produced by the AE and demonstrate comparable performance to those generated by the GAE."
>
> ---
>
> 6. "If I understood correctly, section 2.1 describes the embedding extraction from the GAE applied on the population graph. Maybe the authors could give a different name to the other GAE (the ones for the FC data analysis) because it is confusing (e.g. GAE_FC and GAE_pop) on the figure 1 and in the text."
>
> **Response:**
> Thank you for your suggestions. Following your suggestion, we updated description $GAE_{FC}$ and $GAE_{pop}$ in Figure 1 and highlighted in node features extraction in Method Section.

---

### Author Rebuttal · Authors · 2025-03-08

**Rebuttal:**

We appreciate the opportunity to provide this rebuttal, which allows us to further refine and improve our results. In response to the insightful feedback, we have revised our manuscript to incorporate additional analyses and enhancements aimed at strengthening our conclusions. The updated manuscript has been included in the supporting materials for your review.

And we will respond all reviewers suggesion through official comment.

**Supporting Material:**

/attachment/0c49230be8ac56205b17f788ad93f2f4c9e89bdb.pdf

---

### Meta-Review · Area_Chair_Enn5 · 2025-03-22

**Recommendation:** Accept (Poster)
**Confidence:** 5

**Metareview:**

The most significant review critique was about the relationship between features captured by Grad CAM and behavioral information. The questions have been well answered and appreciated by the reviewer. Although some critiques were related to experimental results were partially addressed, the results look acceptable as a conference paper. Other minor critiques were well addressed, and all reviewers raised their ratings. Authors provide a revised manuscript. Overall, the paper presents an interesting idea to be discussed in MIDL.